# Linking Optimization Success and Stability of Finite-Time Thermodynamics Heat Engines

**DOI:** 10.3390/e27080822

**Published:** 2025-08-02

**Authors:** Julian Gonzalez-Ayala, David Pérez-Gallego, Alejandro Medina, José M. Mateos Roco, Antonio Calvo Hernández, Santiago Velasco, Fernando Angulo-Brown

**Affiliations:** 1Department of Applied Physics, Universidad de Salamanca, 37008 Salamanca, Spain; dpgallego@usal.es (D.P.-G.); amd385@usal.es (A.M.); roco@usal.es (J.M.M.R.); anca@usal.es (A.C.H.); santi@usal.es (S.V.); 2Institute of Physics and Mathematics (IUFFyM), Universidad de Salamanca, 37008 Salamanca, Spain; 3Escuela Superior de Física y Matemáticas, Instituto Politécnico Nacional, Mexico City 07700, Mexico

**Keywords:** finite-time thermodynamics, endoreversible hypothesis, optimization, heat engine stability, thermodynamic success, stochastic perturbations, relative entropy

## Abstract

In celebration of 50 years of the endoreversible Carnot-like heat engine, this work aims to link the thermodynamic success of the irreversible Carnot-like heat engine with the stability dynamics of the engine. This region of success is defined by two extreme configurations in the interaction between heat reservoirs and the working fluid. The first corresponds to a fully reversible limit, and the second one is the fully dissipative limit; in between both limits, the heat exchange between reservoirs and working fluid produces irreversibilities and entropy generation. The distance between these two extremal configurations is minimized, independently of the chosen metric, in the state where the efficiency is half the Carnot efficiency. This boundary encloses the region where irreversibilities dominate or the reversible behavior dominates (region of success). A general stability dynamics is proposed based on the endoreversible nature of the model and the operation parameter in charge of defining the operation regime. For this purpose, the maximum ecological and maximum Omega regimes are considered. The results show that for single perturbations, the dynamics rapidly directs the system towards the success region, and under random perturbations producing stochastic trajectories, the system remains always in this region. The results are contrasted with the case in which no restitution dynamics exist. It is shown that stability allows the system to depart from the original steady state to other states that enhance the system’s performance, which could favor the evolution and specialization of systems in nature and in artificial devices.

## 1. Introduction

This work analyzes the role played by the stability of the auxiliary reservoirs appearing in the endoreversible model, under the key premise that the local equilibrium in the working fluid relaxation can be used as a stability mechanism leading to the steady state of the operation regime. Previous works on the framework of low-dissipation systems have shown that in multiobjective optimization, the Pareto front (the best compromise among a variety of thermodynamic objective functions) coincides with the behavior of the endoreversible model. Thus, at first glance, there is a link between optimization and the endoreversible model even before an actual optimization is carried out. It is known that thermodynamic systems evolve to equilibrium states whenever relaxation times are shorter than the operation or interaction times between the system and its surroundings (reservoirs). In this sense, local equilibrium, responsible for the endoreversible effective behavior of the working fluid (out of equilibrium with external reservoirs), acts as an attractor. In this way, the local equilibrium can define stability dynamics, and it is possible to analyze if the stability, under random perturbations, could point to a self-optimization feature. This point will be tested in the context of *thermodynamic success*.

### 1.1. An Optimization Quest

Since its beginnings, optimization has been a cornerstone of thermodynamics. The role of efficiency in establishing a theoretical upper bound for cyclic energy converters was given in 1824 by Carnot efficiency, 201 years ago [1]. These results represented the first paradigm for heat engine optimization that paved the way in the race for efficiency maximization, where improvements involved the proposal of different geometries in the work and heat (T-S) diagrams aimed at increasing their efficiency and reliability [2,3,4,5]. A second paradigmatic model that accounts for the finite time and finite size of real-life devices was proposed in 1975 by Curzon–Ahlborn (CA) [6], 50 years ago. This model is based on an internally reversible Carnot cycle (endoreversible hypothesis), irreversibly coupled with two external thermal reservoirs. The endoreversible hypothesis has been extensively studied [7,8,9,10,11], and beyond its conceptual idealization, studies on molecular dynamics simulation [12,13] have validated its predictions.

In this model, the coupling between the inner reversible component and the external reservoirs is modeled through heat transfer laws and phenomenological conductances that account for the nature of heat fluxes and the properties of materials involved in heat transport. Initially, it allowed the analysis of the maximum power (MP) regime [6,14,15,16,17,18,19,20,21], and the so-called CA efficiency obtained in this regime, without any doubt, is the main result stemming from the new era of finite-time thermodynamics [22,23,24,25,26,27,28,29]. This efficiency has the form(1)$$\eta =1-\sqrt{\tau },$$
where $\tau =T_{c}/T_{h}$ is the ratio of the external cold and hot heat reservoirs and has proven to be linked to a universal feature of maximum power operation regime [30,31,32,33,34,35,36,37,38]. Over the last five decades, this model and further improvements included the effect of heat leak and internal irreversibilities (affecting the reversible nature of the inner cycle). The model, also denoted as the irreversible Carnot heat engine, has been used to confront a variety of results stemming from macroscopic, mesoscopic, and microscopic scales, allowing the analysis and optimization of several operation regimes and trade-off functions [39,40,41,42,43,44,45,46,47,48,49], leading to similar universal trends for the efficiency at trade-off operation regimes [50,51,52,53].

The aim to provide general aspects related to operation regimes beyond specific models and heat transfer mechanisms is still an active field and new ingredients enter into play, such as the operation parameter control. The latter still has some unsolved issues [54,55] that could be exploited in a general approach of optimization, with possible links to the role of constancy (fluctuations in the energetic output records) in power fluctuations with large efficiencies in quasistatic and steady-state HE models [56,57,58,59]. In this way, issues such as the Carnot efficiency at finite power and efficiency at maximum power have been widely analyzed by different strategies to account for the control of parameters and engine layouts in macroscopic, mesoscopic, and quantum frameworks [56,60,61,62,63,64,65,66,67,68]. All this put to test the idea that *you cannot have it all*, the ultimate quest in thermodynamic optimization, although it is generally accepted that it is impossible to optimize power output, efficiency, and entropy production simultaneously.

### 1.2. Local Equilibrium and Realizations of the Endoreversible Model

The irreversible Carnot-like model has been analyzed analytically through the glasses of mesoscopic heat engines [66], with Otto and Joule–Brayton cycles [4], in the linear irreversible framework [28], as well as in the low dissipation limit [69]. Also, general behaviors typical of the model [70] have been reproduced numerically in simulations of mesoscopic systems [66,67,68]. From a theoretical point of view, the endoreversible nature of the model and the appearance of the auxiliary thermal reservoirs have been understood from the local equilibrium assumptions [28] applied to the working fluid.

In Ref. [12], a 2D finite-time Carnot-like engine was modeled using molecular dynamics simulations of hard-sphere particles confined in a box with a moving piston. In that work, the connection with the linear irreversible model in [28] was proven by analyzing in detail the endoreversible behavior of the gas. For various operating regimes (e.g., maximum power and maximum ecological efficiency), the numerical results matched well with theoretical predictions, especially when internal irreversibility and heat leaks were properly accounted for. The resulting efficiency–power curves were consistent with the endoreversible engine. Unlike idealized models that assume instantaneous adiabats, the simulations revealed the key role of thermal relaxation in finite-time adiabats and isotherms. The statistical behavior of the gas supported an endoreversible-type performance, with near isotherms that were not in equilibrium with the external reservoirs, which can account for the auxiliary reservoirs of the endoreversible model. For completeness, in Figure 1 the results reported in [12] are depicted. In a similar dynamical simulation model, some results of the linear response regime were recovered [13].

These results point to the fundamental role played by the dynamics between the working fluid and reservoirs in finite-size and finite-time systems [71,72,73,74,75,76]. This feature provides a first glance at the connection between entropy generation (irreversibilities), working fluid–reservoir correlations, and energetic success, discussed in Section 3. This is also in line with previous efforts to provide a more extended vision of finite-time thermodynamics to include a dynamical structure (involving time or rate constraints) with the theory and add entropy-related requirements to the optimization of operation regimes [77,78,79,80,81].

### 1.3. Optimization Due to Stability

A series of papers [82,83,84,85] explored how a stability mechanism that maintains a heat engine operating under steady-state conditions could induce an energetic optimization. The main idea is to consider that the constancy of the heat fluxes or the energetic function that defines the operation regime is responsible for maintaining the operation variables in the steady state. First, this was analyzed for heat engines and refrigerators in the low-dissipation model [83,84], consistent with the irreversible Carnot-like and the endoreversible models. Later, this feature was analyzed for the endoreversible model [85]. In [83,84] it was shown that the relaxation trajectories to the steady state exhibit an optimization process in power output, efficiency, and entropy generation. This was also proven for consecutive random perturbations on the system, simulating external perturbations that could displace the operation state from the steady state. Statistically, it was shown that the deviation of the operation state is displaced in the direction of the Pareto front of the system, which coincides with the endoreversible limit and in the direction of larger efficiencies and lower entropy generation, with a slight sacrifice in power output. When this behavior was explored in the endoreversible model, the self-optimization was not as evident as in the low-dissipation case.

An interesting subject that has not yet been analyzed is what happens when the stability mechanism that keeps the system in a steady state is linked to the local equilibrium that can effectively reproduce the behavior of the endoreversible model. One major goal of studying the lack of control, is to shed some light on “natural optimization preferences” [86,87,88,89,90,91] in evolution and adaptability to the environment [92,93].

The present paper is organized as follows. In Section 2, the irreversible Carnot-like heat engine (an extension of the endoreversible model) and the operation regimes of maximum power, efficiency, and ecological and Ω functions are addressed. Section 3 addresses a metric to evaluate the success of energy converters. In Section 4, the stability of heat reservoirs (linked to the internal irreversibility) is discussed to justify using the irreversibility of the internal cycle in the definition of stability dynamics. Section 6 presents the energetic trajectories of the system in the relaxation to the steady state. In Section 7 the effect of stochastic perturbations and the evolution of the operation regime induced by the stability is analyzed. Finally, some concluding remarks are discussed.

## 2. Irreversible Carnot-like Heat Engine

For self-contained purposes, the mathematical model of an irreversible Carnot-like heat engine is summarized. This model is an extension of the original Curzon–Ahlborn endoreversible heat engine proposed in [6] and the simplified versions introduced in [17,19,20,76]. It represent a standard model widely used in finite-time thermodynamics [94]. The endoreversible hypothesis assumes an internal Carnot cycle operating between the temperatures $T_{h}^{\prime }$ and $T_{c}^{\prime }\leq T_{h}^{\prime }$. This Carnot cycle is coupled irreversibly to the external heat baths at temperatures $T_{h}$ and $T_{c}\leq T_{c}^{\prime }\leq T_{h}^{\prime }\leq T_{h}$, and an external heat leak between these two external heat reservoirs is accounted for. The input and output heat fluxes, ${\dot{Q}}_{h}$ and ${\dot{Q}}_{c}$, as well as the heat leak ${\dot{Q}}_{L}$, obey the Newton heat transfer law(2)$$\begin{array}{c} |{\dot{Q}}_{h}|=\sigma_{h}\left(T_{h}-T_{h}^{\prime }\right)\equiv \sigma_{h}T_{h}\left(1-\frac{1}{a_{h}}\right), \end{array}$$(3)$$\begin{array}{c} |{\dot{Q}}_{c}|=\sigma_{c}\left(T_{c}^{\prime }-T_{c}\right)\equiv \sigma_{c}T_{c}\left(a_{c}-1\right), \end{array}$$(4)$$\begin{array}{c} |{\dot{Q}}_{L}|=\sigma_{i}\left(T_{h}-T_{c}\right)\equiv \sigma_{i}T_{h}\left(1-\tau \right), \end{array}$$
where $a_{h}=T_{h}/T_{h}^{\prime }\in \left[1,\tau^{-1}\right]$, and $a_{c}=T_{c}^{\prime }/T_{c}\in \left[1,\tau^{-1}a_{h}^{-1}\right]$; the $\sigma_{c}$, $\sigma_{h}$ and $\sigma_{i}$ are the corresponding thermal conductances, which are considered as constant, and $t_{c}$ and $t_{h}$ are the operation times. In the endoreversible engine, the entropy generation due to the heat enchanged in the isotherms lead to the Clausius equality, $|{\dot{Q}}_{c}|/T_{c}^{\prime }-\left|{\dot{Q}}_{h}\right|/T_{h}^{\prime }=0$. However, to allow for internal irreversibilities, the irreversibility parameter I≤1 is introduced. This parameter compensates the larger entropy produced in the colder isotherm (or a deficit in the hot reservoir) so the engine fulfills the Clausius equality,(5)$$I\frac{|{\dot{Q}}_{c}|}{T_{c}^{\prime }}-\frac{|{\dot{Q}}_{h}|}{T_{h}^{\prime }}=0,$$

Thus, it provides a measure of the deviation from the reversible hypothesis for the working fluid. According to this definition, the inner cycle is reversible in the case I=1 and irreversible if I<1. The internal temperatures $T_{h}^{\prime }$ and $T_{c}^{\prime }$ (i.e., $a_{h}$ and $a_{c}$) are not independent. From Equations (Equation 2), (Equation 3) and (Equation 5),(6)$$a_{c}=\frac{I}{I-\sigma_{hc}\left(a_{h}-1\right)},$$
where $\sigma_{hc}\equiv \sigma_{h}/\sigma_{c}$. From the above equations, the power and the efficiency are given by(7)$$\begin{array}{ccc} P= & \;|{\dot{Q}}_{h}\left|-\right|{\dot{Q}}_{c}|\; & =\sigma_{h}T_{h}\left(1-a_{h}^{-1}-\frac{\tau \left(a_{h}-1\right)}{I+\sigma_{hc}a_{h}}\right),\; \end{array}$$(8)$$\begin{array}{ccc} \eta = & 1-\frac{|{\dot{Q}}_{c}\left|+\right|{\dot{Q}}_{L}|}{|{\dot{Q}}_{h}\left|+\right|{\dot{Q}}_{L}|} & =\left(1-\frac{a_{h}\tau }{I-\sigma_{hc}\left(a_{h}-1\right)}\right)\left(\frac{a_{h}-1}{a_{h}-1+\sigma_{ih}a_{h}\left(1-\tau \right)}\right), \end{array}$$
where $\sigma_{ih}\equiv \sigma_{i}/\sigma_{h}$. The Curzon–Ahlborn endoreversible model is recovered if I=1 and $\sigma_{i}=0$ [6]. The entropy production is given by(9)$$\begin{array}{c} \sigma =\frac{|{\dot{Q}}_{c}|}{T_{c}}-\frac{|{\dot{Q}}_{h}|}{T_{h}}+\left|{\dot{Q}}_{L}\right|\left(\frac{1}{T_{c}}-\frac{1}{T_{h}}\right)=\sigma_{h}\left(\frac{1}{a_{h}}-1+\frac{a_{h}-1}{I-\sigma_{hc}\left(a_{h}-1\right)}+\frac{\sigma_{ih}{\left(1-\tau \right)}^{2}}{\tau }\right), \end{array}$$

### Maximizing Power, Efficiency, Ecological and Omega Figures of Merit

The power output is a convex function that exhibits a maximum only in the direction of $a_{h}$. The analytical maximum stemming from the condition $\frac{dP}{da_{h}}=0$ gives the value(10)$$a_{h,MP}=\frac{I+\sigma_{hc}}{\sigma_{hc}+\sqrt{I\tau }},$$
producing an efficiency and power output(11)$$\begin{array}{c} \eta_{MP}=\frac{{\left(\sqrt{I}-\sqrt{\tau }\right)}^{2}}{I-\sqrt{I\tau }+\sigma_{ih}\left(\sigma_{hc}+I\right)\left(1-\tau \right)}, \end{array}$$(12)$$\begin{array}{c} \;P_{MP}=\frac{{\left(\sqrt{I}-\sqrt{\tau }\right)}^{2}}{I+\sigma_{hc}}, \end{array}$$
which, in the limit I→1 and $\sigma_{ih}\to 0$, leads to the well-known Curzon–Ahlborn efficiency, $\eta_{CA}$ (which is independent of the conductances) and the maximum power given by(13)$$\begin{array}{c} \eta_{MP}=\eta_{CA}=1-\sqrt{\tau }, \end{array}$$(14)$$\begin{array}{c} \;P_{MP}=\frac{{\left(1-\sqrt{\tau }\right)}^{2}}{1+\sigma_{hc}}. \end{array}$$

This efficiency is linked with a universal feature of the maximum work and maximum power regimes. The approximation to the first order in the Taylor series in terms of $\eta_{C}$, $\eta_{CA}=\frac{\eta_{C}}{2}+\frac{\eta_{C}^{2}}{8}+...$ represents a lower bound for the maximum power regime for the low-dissipation regime and for the phenomenological heat transfer law. As it will be shown later, this value, $\frac{\eta_{C}}{2}$, is meaningful in describing the success of energy conversion.

The corresponding maximization of the efficiency gives the $a_{h,M\eta }$ value(15)$$a_{h,M\eta }=1-\frac{\sigma_{ih}\left(1-\tau \right)\left(I\left(\tau -\sqrt{I\tau }\right)+\sigma_{hc}\left(I-\sqrt{I\tau }\right)\right)-\sqrt{I\sigma_{ih}\tau \left(I-\tau \right)\left(1-\tau \right)\left(I+\sigma_{hc}\right)}}{\tau \left(\sigma_{hc}+I\right)+\sigma_{ih}\left(I\tau -\sigma_{hc}^{2}\right)\left(1-\tau \right)},$$
that for zero heat leak leads to $a_{h,M\eta }=1$, since the efficiency is a monotonous decreasing function of $a_{h}$.

The ecological function, *E*, and the Omega function, Ω, are between the so-called trade-off figures of merit. It is noteworthy that both functions differ in their interpretation, but for the irreversible Carnot engine, both lead to the same expression [49]. The ecological function was first introduced in Ref. [39] and refers to a compromise between the power output and the entropy generation released to the cold external reservoir(16)$$E\equiv P-T_{c}\sigma =\left({\dot{Q}}_{h}+{\dot{Q}}_{L}\right)\left(2\eta -\eta_{C}\right),$$
while the Ω function is a trade-off between power loss and power gained when compared with a given power output and with fixed heat input, that is,(17)$$\Omega \equiv P_{gain}-P_{loss}=\left(P-P_{min}\right)-\left(P-P_{max}\right)=2P-\left({\dot{Q}}_{h}+{\dot{Q}}_{L}\right)\eta_{C}=\left({\dot{Q}}_{h}+{\dot{Q}}_{L}\right)\left(2\eta -\eta_{C}\right).$$

For the present model, it has the form(18)$$\Omega =E=\sigma_{h}T_{h}\left(\left(1+\tau \right)\left(1-a_{h}^{-1}\right)-\sigma_{ih}{\left(1-\tau \right)}^{2}+\frac{2\tau \left(a_{h}-1\right)}{\sigma_{hc}\left(a_{h}-1\right)-I}\right).$$

Its maximum value is reached at(19)$$a_{h,ME}=a_{h,M\Omega }=\frac{I+\sigma_{hc}}{\sigma_{hc}+\sqrt{\frac{2I\tau }{1+\tau }}},$$
which, in the case of no heat leak, gives the value(20)$$\eta_{ME}=\eta_{M\Omega }=1-\sqrt{\frac{\tau \left(1+\tau \right)}{2I}},$$
and for the endoreversible case, I=1, a well-known ecological efficiency is recovered; this efficiency also has been related to a universal feature of the ecological and the Ω regimes, whose linear term of the Taylor series is in terms of $\eta_{C}$, $\eta_{E}=\frac{3\eta_{C}}{4}+\frac{\eta_{C}^{2}}{32}+...$, and also represents a lower bound for the maximum ecological and Omega regimes [49].

Beyond trade-off operation regimes, it is possible to analyze the success of any energy converter operating in cycles. This will be addressed in the following section.

## 3. Losses and Success of Energy Converters

In [75], stability criteria for the heat reservoirs were defined by the states of minimum internal energy, $U_{min}$, and maximum entropy, $S_{max}$, which is analogous to the evolution of closed systems towards equilibrium states. In the stationary operation state, as long as the reservoirs are exchanging heat irreversibly with the working fluid, an inherent entropy generation due to the correlations among them appears. Details on this correlation entropy generation are addressed later in Section 4. These extremal stable states are obtained in the reversible limit, where the changes in internal energy and entropy of a thermal reservoir are only caused by heat exchanges.(21)$$\begin{array}{ccc} \Delta U & \to & U\left(t\right)=U_{0}-\left|Q_{h}\left(t\right)\right|\eta \leq 0,\;\;\;\;\;\;\;\;\; \end{array}$$(22)$$\begin{array}{ccc} \Delta S & \to & S\left(t\right)=S_{0}+\frac{|Q_{h}\left(t\right)|}{T_{h}}\frac{\eta_{C}-\eta }{1-\eta_{C}}\geq 0, \end{array}$$
where both are monotonous functions of efficiency. When the efficiency is the highest ($\eta_{C}$), $U_{min}\left(t\right)$ is obtained, and at the minimum efficiency, η=W(t)=0 (all the heat is dissipated), the value of $S_{max}\left(t\right)$ is recovered. Two parameters that provide a measure of the nearness to the extremal situations $U_{min}$ and $S_{max}$ are defined by(23)$$\begin{array}{ccc} E_{U} & \equiv & \frac{U(t)}{U_{min}\left(t\right)}=\frac{\eta }{\eta_{C}},\;\;\;\;\;\;\; \end{array}$$(24)$$\begin{array}{ccc} E_{S} & \equiv & \frac{S(t)}{S_{max}\left(t\right)}=\frac{\eta_{C}-\eta }{\eta_{C}}, \end{array}$$
fulfilling that $E_{U}\leq 1$ and $E_{S}\leq 1$. The Euclidean distance between $E_{U}$ and $E_{S}$ can be computed through the *p*–norm(25)$$N_{p}\equiv {\left(|E_{U}{|}^{p}+{\left|E_{S}\right|}^{p}\right)}^{1/p},$$
which is a concave function with one minimum at $\eta^{*}=\frac{\eta_{C}}{2}$ except for p=1 (Taxicab or Manhattan distance). Thus, at $\eta^{*}=\frac{\eta_{C}}{2}$, the two extreme configurations are at the minimum distance. This means that $\eta^{*}$ represents a threshold from which the operation state increases the distance between the extremal stable states given by the equilibrium of the reservoir.

In the general analysis presented in [75], an open question remained regarding whether one would expect that the evolution of energy converters will tend to favor optimum energetic performance. In such a case, reversibility (linked to reservoir stability) is another ingredient to consider together with the operation regime. In this context, three quantities are introduced to measure the success of a system, the parameters(26)$$\begin{array}{ccc} R & \equiv & \frac{\eta -\eta_{min}}{\eta_{max}-\eta_{min}}, \end{array}$$(27)$$\begin{array}{ccc} D & \equiv & \frac{\Delta S_{Univ}}{\Delta S_{Univ}^{max}},\;\;\;\;\;\;\;\; \end{array}$$
where the reversibility coefficient, R∈(0,1), indicates how close the efficiency is to the maximum efficiency, and the irreversibility coefficient, D∈(0,1), indicates the nearness to the maximum entropy generation state. $\eta_{max}=\eta_{C}$, $\eta_{min}=0$, and the entropy change in the thermodynamic universe is(28)$$\Delta S_{univ}=-\frac{|Q_{h}|}{T_{h}}+\frac{|Q_{c}|}{T_{c}}=\frac{|Q_{h}|}{T_{h}}\frac{\eta_{C}-\eta }{1-\eta_{C}}\geq 0,$$
which has a maximum when no work is delivered, i.e., η=0. Thus, $\Delta S_{univ}^{max}=\frac{|Q_{h}|}{T_{h}}\frac{\eta_{C}}{1-\eta_{c}}$. These definitions of *R* and *D* coincide with $E_{U}$ and $E_{S}$ (Equations (Equation 23) and (Equation 24)),(29)$$\begin{array}{ccc} R & = & \frac{\eta }{\eta_{C}}=E_{U},\;\;\; \end{array}$$(30)$$\begin{array}{ccc} D & = & \frac{\eta_{C}-\eta }{\eta_{C}}=1-\frac{\eta }{\eta_{C}}=1-R=E_{S}, \end{array}$$
respectively, and the difference, which indicates if the system is closer to the state of maximum efficiency or closer to the maximum entropy generation state,(31)$$R-D=2R-1=\frac{2\eta }{\eta_{C}}-1.$$
has a zero at $\eta^{*}=\frac{\eta_{c}}{2}$. The case R>D defines the *region of success since the reversibility dominates over irreversibility, while the opposite condition defines the* failure region. Thus, $\eta^{*}=\frac{\eta_{C}}{2}$ should be considered as the minimum value of the efficiency of any thermodynamically successful heat engine, which interestingly, is also the lower bound for the efficiency in the maximum power operation regime and where the distance to stable states for reservoirs ($E_{U}=1$ and $E_{S}=1$) is minimum.

In [76], the parameter *D* was decomposed into external losses, $D_{e}$, and internal losses, $D_{i}$. For the irreversible Carnot-like engine, these are defined as(32)$$\begin{array}{ccc} D_{e} & = & \frac{\eta_{c}-\eta_{endo}}{\eta_{c}}=1-\frac{\eta_{endo}}{\eta_{c}}, \end{array}$$(33)$$\begin{array}{ccc} D_{i} & = & \frac{\eta_{endo}-\eta }{\eta_{endo}}=1-\frac{\eta }{\eta_{endo}},\; \end{array}$$
which gives the deviation of the endoreversible engine from the Carnot cycle and the deviation of the actual engine from the endoreversible one, respectively. These two parameters are linked with *D* as follows:(34)$$\begin{array}{ccc} D & \equiv & \frac{\eta_{C}-\eta }{\eta_{C}}=\frac{\eta_{C}-\eta_{endo}}{\eta_{C}}+\frac{\eta_{endo}-\eta }{\eta_{endo}}\left(\frac{\eta_{endo}}{\eta_{C}}\right)\leq 1, \end{array}$$(35)$$\begin{array}{ccc} D & = & D_{e}+\left(\frac{\eta_{endo}}{\eta_{C}}\right)D_{i}=D_{e}+D_{i}-D_{e}D_{i}.\;\;\;\;\;\;\;\;\;\;\;\;\;\;\;\;\;\;\; \end{array}$$

The values I<1 and $\sigma_{i}\neq 0$ will provide information on the internal and external dissipations. Then, both loss factors are computed as(36)$$\begin{array}{ccc} D_{e} & = & 1-\frac{\eta (a_{h},I=1,\sigma_{hc},\sigma_{ih}=0,\tau )}{1-\tau }=1-\frac{\left(1-\frac{a_{h}\tau }{1-\left(a_{h}-1\right)\sigma_{hc}}\right)}{1-\tau },\;\;\;\;\;\;\;\;\;\;\;\;\;\;\;\;\;\;\;\;\;\;\;\;\;\; \end{array}$$(37)$$\begin{array}{ccc} D_{i} & = & 1-\frac{\eta (a_{h},I,\sigma_{hc},\sigma_{ih},\tau )}{\eta (a_{h},I=1,\sigma_{hc},\sigma_{ih}=0,\tau )}=1-\frac{1+\frac{a_{h}\tau }{\sigma_{hc}\left(a_{h}-1\right)-I}}{\left(1+\frac{a_{h}\tau }{\sigma_{hc}\left(a_{h}-1\right)-1}\right)\left(1+\frac{a_{h}\sigma_{ih}\left(1-\tau \right)}{a_{h}-1}\right)}. \end{array}$$

With these parameters and the reversible and irreversible coefficients, *R* and *D*, it is possible to evaluate the effect of instabilities and the energetic evolution of the operation regime given by the relaxation mechanism towards stability. In the following, an analysis to connect stability and success will be presented.

## 4. Heat Reservoirs Stability

Intuition dictates that the natural evolution of cyclic energy converters will tend to favor optimum energetic performance. Thus, if stable points are attraction configurations, those that tend to null performance are doomed to vanish, and those leaning to optimum efficiency will present a trade-off with the actual needs of the energy conversion and the stable conditions of the system acting as a heat source. This point can be explored by analyzing stability dynamics and the effect of perturbations on the system can be studied.

The core of the endoreversible hypothesis lies in the definition of the working fluid isotherms and the non-equilibrium interaction with the external reservoirs, whose stability is compromised by internal irreversibilities. Thus, when departing from reversible exchanges of heat between the reservoirs and the working fluid, the relaxation mechanism in both components is crucial. A good approximation to reversible heat sources requires that relaxation times are sufficiently short compared to operation times. Heat transfers between the working fluid and the reservoirs present unavoidable consequences on their internal modes, and the entropy of the compound system (system+reservoirs) exhibits correlations between them; this is a source of instabilities for the heat reservoirs. This subject has been cleverly addressed in [74] by analyzing the link with an irreversible contribution to the entropy change of a system. From a quantum framework, entropy production can be linked directly to a measure of the correlation/entanglement between the system and reservoirs. Here, the focus is on the macroscopic framework.

The total entropy of the compound system is not only the sum of the entropy of the reservoirs, $S_{r}$, and the system, $S_{s}$, but there is an additional term due to the correlations between them, $S_{c}$, in such a way that(38)$$S=S_{s}+\sum_{r}S_{r}+S_{c}.$$

Under initial equilibrium conditions, the reservoirs do not present correlations, and thus $S_{c}\left(0\right)=0$ (at time t=0). For further times, a coupling is in place, and there is an irreversible contribution to the entropy change of the compound system. This term is measured as the relative entropy between the actual state of the compound system and that when the heat reservoirs are in thermal equilibrium with no correlations between the system and the reservoirs, that is(39)$$\begin{array}{c} \Delta_{i}S\left(t\right)=D\left[P\left(t\right)∥P_{s}\left(t\right)\prod_{r}P_{r}^{eq}\left(t\right)\right]=P\left(t\right)ln\left(\frac{P(t)}{P_{s}\left(t\right)\prod_{r}P_{r}^{eq}\left(t\right)}\right), \end{array}$$
where $D\left[\cdot ∥\cdot \right]$ is the relative entropy, or Kullback–Leibler divergence, $P_{s}$ is the probability for the system, $P_{r}^{eq}$ is that for the *r*-th reservoir at equilibrium conditions and *P* the actual probability for the system+reservoirs. This total entropy change of the system, $\Delta_{i}S\left(t\right)$, contains at least the irreversible entropy contribution of the correlations, $\Delta_{i}S\left(t\right)\geq -\Delta S_{c}\left(t\right)=-S_{c}\left(t\right)\geq 0$, and fulfills the inequality [74](40)$$\Delta_{i}S\left(t\right)\geq -S_{c}\left(t\right)=\Delta S_{s}\left(t\right)+\sum_{r}\Delta S_{r}\left(t\right)\geq 0.$$

In the case of one reservoir, a version of the maximum work theorem is obtained,(41)$$T\Delta_{i}S\left(t\right)=W\left(t\right)-\Delta F_{s}\left(t\right)\geq 0,$$
where $F_{s}$ is the free energy of the system, and $T\Delta_{i}S\left(t\right)=0$ is only for reversible engines.

In the dynamics simulation (Section 1.2), the term associated with the correlations is hidden in the random momentum change in each collision between the particles and the thermalizing wall. However, from Equation (Equation 41) for the isothermal process at temperature $T_{c}^{\prime }$, and using the Claussius relation given in Equation (Equation 5), the irreversible contribution to the entropy change can be linked with the parameter *I*, as follows: (42)$$\begin{array}{ccc} T_{c}^{\prime }\Delta_{i}S & = & W-\Delta F_{s}=\Delta U-Q_{c}-\Delta U+T_{c}^{\prime }\Delta S_{T_{c}^{\prime }}=-Q_{c}+T_{c}^{\prime }\Delta S_{T_{c}^{\prime }}\geq 0, \end{array}$$(43)$$\begin{array}{ccc} \Rightarrow \Delta_{i}S & = & \frac{|Q_{c}|}{T_{c}^{\prime }}+\Delta S_{T_{c}^{\prime }}=\frac{|Q_{c}|}{T_{c}^{\prime }}-I\frac{|Q_{c}|}{T_{c}^{\prime }}=\left(1-I\right)\frac{|Q_{c}|}{T_{c}^{\prime }}\geq 0.\;\;\;\;\;\;\;\;\;\;\;\;\;\;\;\;\;\;\;\;\; \end{array}$$

In this way, $\Delta_{i}S$ is interpreted as the irreversible entropy that is compensated by the parameter *I* so that the system fulfills the Clausius inequality in Equation (Equation 5).

Returning to the expressions of $E_{U}$ and $E_{S}$ (Equations (Equation 23) and (Equation 24)), notice that they coincide with *R* and *D* (Equations (Equation 29) and (Equation 30)), respectively. The configurations where $E_{U},R\to 1$ can only be achieved under totally reversible conditions, for which Equations (Equation 41) and (Equation 43) are zero and reservoirs are in equilibrium with no correlations with the system. The other zero for $\Delta S_{i}\left(t\right)$ is found in the opposite case, where $E_{S},D\to 1$ since W(t)=0. Thus, the input heat is directly dissipated into the external cold reservoir at temperature $T_{c}$, and no correlations take place between the system and the external reservoirs; in that case,$$\Delta S=\sum_{r}\frac{Q_{r}}{T_{r}}.$$

These two extreme situations are incompatible with real applications, for one has no use, and the other is linked to zero power. In real situations, there are indeed correlation dynamics between reservoirs and the system that can lead ultimately to instabilities of the heat reservoirs. The previous discussion provides a justification to define stability dynamics where the internal irreversibility of the system enters into play. In the finite-time thermodynamics phenomenological model, it is not possible to properly account for the mentioned correlations. However, the parameter *I* is linked with $\Delta S_{i}\left(t\right)$, which contains the entropy generated in the correlations (linked with the stability of the heat reservoirs), justifying the use of *I* as a dynamical quantity involved in the definition of a stability dynamics. This will be addressed in the following section.

## 5. Stability Dynamics

A stability dynamics that contains information about the operation regime and the stability of the heat reservoirs requires at least two dynamical equations. For this analysis, two functions will be used to define such stability. One will be the Ω function under MΩ conditions, determined by the control parameter $a_{h}$. The other one is the relaxation of the working fluid, responsible for fulfilling the endoreversible hypothesis, and measured by the irreversibility parameter *I*. By assuming that the MΩ regime is a steady state, then the most general stability dynamics for small perturbations follows the linear approximation; that is,(44)$$\left(\begin{array}{c} \frac{dI}{dt}\\ \frac{da_{h}}{dt} \end{array}\right)=-\left(\begin{array}{cc} C_{I} & 0\\ 0 & C_{a_{h}} \end{array}\right)\left(\begin{array}{c} I-I^{M\Omega }\\ a_{h}-a_{h}^{M\Omega } \end{array}\right),$$
which gives the evolution in time, *t*, of the parameters *I* and $a_{h}$, and $C_{I}$ and $C_{a_{h}}$ give the restitution strength in the corresponding variables, which, with the information at hand, cannot be determined, and different scenarios for them will be addressed in the following. These two parameters define the internal working temperature $T_{c}^{\prime }$ through $a_{c}=a_{c}\left(a_{h},I\right)$ (Equation (Equation 6)) and the Ω function $\Omega \left(a_{h},I\right)$ (Equations (Equation 16) and (Equation 17)). The Taylor expansion to linear order of these two quantities gives(45)$$\left(\begin{array}{c} a_{c}-a_{c}^{M\Omega }\\ \Omega -\Omega^{M\Omega } \end{array}\right)=J\left(\begin{array}{c} I-I^{M\Omega }\\ a_{h}-a_{h}^{M\Omega } \end{array}\right)=\left(\begin{array}{cc} {\left.\frac{\partial a_{c}}{\partial I}\right|}_{M\Omega } & {\left.\frac{\partial a_{c}}{\partial a_{h}}\right|}_{M\Omega }\\ {\left.\frac{\partial \Omega }{\partial I}\right|}_{M\Omega } & {\left.\frac{\partial \Omega }{\partial a_{h}}\right|}_{M\Omega } \end{array}\right)\cdot \left(\begin{array}{c} I-I^{M\Omega }\\ a_{h}-a_{h}^{M\Omega } \end{array}\right)$$
and from Equations (Equation 44) and (Equation 45), the final expression for the linear stability dynamics can be written as(46)$$\left(\begin{array}{c} \frac{dI}{dt}\\ \frac{da_{h}}{dt} \end{array}\right)=-\left(\begin{array}{cc} C_{I} & 0\\ 0 & C_{a_{h}} \end{array}\right){\left(\begin{array}{cc} {\left.\frac{\partial a_{c}}{\partial I}\right|}_{M\Omega } & {\left.\frac{\partial a_{c}}{\partial a_{h}}\right|}_{M\Omega }\\ {\left.\frac{\partial \Omega }{\partial I}\right|}_{M\Omega } & {\left.\frac{\partial \Omega }{\partial a_{h}}\right|}_{M\Omega } \end{array}\right)}^{-1}\left(\begin{array}{c} a_{c}-a_{c}^{MP}\\ \Omega -\Omega^{M\Omega } \end{array}\right),$$
resulting in the dynamics equations(47)$$\begin{array}{ccc} \frac{dI}{dt} & = & g\left(a_{h},I;C_{a_{h}},C_{I}\right), \end{array}$$(48)$$\begin{array}{ccc} \frac{da_{h}}{dt} & = & h\left(a_{h},I;C_{a_{h}},C_{I}\right). \end{array}$$

This dynamics in the *I*-$a_{h}$ space is depicted in Figure 2. The values $\sigma_{hc}=1.4$, $\sigma_{ih}=0.2$, $I_{M\Omega }=0.9$, and τ=0.5 are used as a representative configuration as they approximate the behavior of the irreversible Carnot-like heat engine to the one obtained in the simulation shown in Figure 1. The value $a_{h,M\Omega }$ is calculated from Equation (Equation 19). Figure 2a represents the case where the dynamics in the $a_{h}$ direction is stronger, in Figure 2b, where the restitution strength $C_{I}$ and $C_{a_{h}}$ are symmetrical, while in Figure 2c the dynamics in $C_{I}$ is the strongest. The streamlines show a trend of the relaxation trajectories to increase the parameter *I*, making the engine closer to the endoreversible limit.

After a perturbation, the relaxation trajectories evolve with a variable speed(49)$$v=\sqrt{{\left(dI/dt\right)}^{2}+{\left(da_{h}/dt\right)}^{2}},$$
which is depicted in Figure 3 (left). The influence of the relaxation speed is significant since slow relaxations under many perturbations will effectively modify the operation state. In Figure 3 (right), representative relaxation trajectories starting from conditions near zero power output provide a general idea of how other initial configurations will evolve in time. Also, the states of maximum power (MP, red dashed straight line), the maximum ecological and Omega functions (MΩ, green dashed line), and the maximum efficiency (Mη, blue dashed line) are indicated. All the trajectories will finally arrive at the steady state (MΩ). However, for a given relaxation time, the trajectories that arrive at the steady state are colored in red, and those that require longer times are depicted in light yellow. For this representation, the time defined is a multiple of the characteristic relaxation time, $t_{relax}$, which is calculated from the eigenvalues of the dynamics matrix appearing in Equation (Equation 46). For the representation of Figure 3 (right), a time of $11t_{relax}$ is sufficient to analyze fast and slow relaxation trajectories. Notice in Figure 3 (right) that the states of maximum efficiency are in a region of slow velocity, and, in comparison, the states of maximum power evolve with higher velocities.

In the context of success in the performance of heat devices, it is interesting to keep an eye on the location of the region where the reversible parameter R≥1/2, which is where $\eta \geq \eta_{C}/2$. This is also depicted in Figure 3 (right). It is noteworthy that the velocities that take the system out of the unsuccessful area are greater and once the system is in the region of success, the return velocities are slower. Also, the MP states are in the success region only for values of *I* close to 1, while for Mη and MΩ and ME, the region of success covers a more extended area. In all this area, Ω>0 and its maximum are located between the MP and Mη states.

## 6. Energetic Evolution in the Relaxation Towards the Steady State

The relaxation trajectories strongly depend on the restitution strength in the direction of *I* or $a_{h}$, which can be analyzed through the restitution coefficients $C_{I}$ and $C_{a_{h}}$. The velocity is shown in Figure 4 for the three cases of evolution: for the dynamics being dominated more strongly by *I*, for the symmetrical case $C_{I}=C_{a_{h}}=1$, or where the dynamics are dominated by $a_{h}$, this is depicted in Figure 4a, b and c, respectively. Notice that larger velocities are produced when *I* dominates the dynamics. This behavior is expected if the relaxation time leading to local equilibrium and the effective isothermal processes for the working fluid is considerably smaller than the operation time. On the other hand, when the dynamics are more strongly given by the operation variable $a_{h}$, the convergence to $a_{h,M\Omega }$ is fast, but afterward, the evolution is noticeably slow.

The relaxation trajectories for the three cases depicted in Figure 4 are shown in Figure 5. In all the cases, it is relevant that the trajectories rapidly enter into the zone where R>1/2, the so-called “success region”. In this region, the MΩ states are located between the MP and Mη ones and the curves approaching the MP are moving faster than those moving between the MΩ and Mη regimes. For illustrative purposes, the trajectories evolving with decreasing values of $a_{h}$ and are slower and indicated with dashed orange lines, which can be tracked in the forthcoming figures. These trajectories start from points of low or zero power output.

Figure 6 shows the trajectories in the energetic spaces P−η and σ−η. The blue-shaded region corresponds to the zone where R<1/2 (or $\eta <\eta_{C}/2$). All the trajectories that exit the shaded region to enter into the success region in the P−η space simultaneously increase power and efficiency, while the trajectories appearing in Figure 5 are evolving from the right exit of the shaded region with trajectories that also decrease entropy generation. Those that evolve by increasing entropy generation tend to do so with small increases in σ.

The evolution of the reversible coefficient *R* with respect to $a_{h}$ and *I* is shown in Figure 7 in the first and second rows, respectively. The cases depicted correspond to those presented in Figure 5. Meanwhile, in the first row, the trajectories can exit the non-success region by increasing $a_{h}$ (short trajectories) or decreasing $a_{h}$ (with long and rapid trajectories). In the second row, all the trajectories evolve in the shaded region with trajectories that increase the value of *I*. There are inflection points from which the system cannot continue becoming more reversible, and the system has to converge to the steady state by decreasing *R*. These trajectories seem to focus on the simultaneous improvement of *R* and *I* even though, later, the system has to return to converge to the MΩ state.

The evolution of the components $D_{e}$ (Equation (Equation 36)) and $D_{i}$ (Equation (Equation 37)) corresponding to the external and internal losses is depicted in Figure 8. The total loss parameter *D* (Equation (Equation 35)) also gives information regarding the success region, which corresponds to the region where D<1/2. Overall, the curves tend to exhibit larger variations in the value of $D_{e}$ when this parameter has to decrease; otherwise, the increase in $D_{e}$ is quite small. In any case, the evolution curves always tend to quickly decrease $D_{i}$ to move to the success region.

All these cases show a clear influence of the stability dynamics in the way the system relaxes to a steady state. Optimum states given by locus near maximum efficiency and maximum trade-off functions such as the ecological and Omega functions produce slower relaxation dynamics, while trajectories passing outside the optimization region of interest exhibit faster relaxation endings to simultaneously improve power, efficiency, and entropy production while improving the reversibility of the system and decreasing the internal irreversibility parameter $D_{i}$.

## 7. Consecutive Perturbations

### 7.1. Stochastic Trajectories

The previous analysis on a single trajectory after a perturbation has indicated the possible role of stability in enhancing the system operation state. However, in realistic conditions, it is expected that heat devices operating under stationary operation conditions experience small fluctuations produced from external factors and from the intrinsic variability due to limited control. Now, it is time to analyze the effect of consecutive perturbations over one cycle. These perturbations will be modeled by stochastic variations in both parameters, $a_{h}$ and *I*, and test if the relaxation dynamics, as would be expected, tend to push the operation regime to more optimum states in terms of power, efficiency, and entropy generation. Based on the stability dynamics given by Equations (Equation 47) and (Equation 48), stochastic perturbations can be incorporated as an additive white noise with two normally distributed random variables. The independent stochastic variables $\left\{\xi_{1},\;\xi_{2}\right\}$ in the *I*-$a_{h}$ directions obey a two-dimensional Gaussian distribution(50)$$f_{\xi }\left(I,a_{h}\right)=\frac{\beta^{2}}{\pi \;I_{M\Omega }\;a_{h,M\Omega }}\;e^{-\beta^{2}\left[{\left(\frac{I}{2I_{M\Omega }}\right)}^{2}+{\left(\frac{a_{h}}{2a_{h,M\Omega }}\right)}^{2}\right]},$$
where β=40 is a good compromise to maintain the system around the steady state for long periods. The standard deviations are proportional to the control variables, $\sigma_{I}=I_{M\Omega }/\sqrt{2}\beta \approx 0.018\;I_{M\Omega }$ and $\sigma_{a_{h}}=a_{h,M\Omega }/\sqrt{2}\beta \approx 0.018a_{h,M\Omega }$. Based on Equations (Equation 47) and (Equation 48), the corresponding stochastic differential equations that numerically describe the system evolution are solved using the Euler–Maruyama method [95].(51)$$\begin{array}{ccc} \Delta I & = & g\left(a_{h},I;C_{a_{h}},C_{I}\right)\Delta t+\xi_{1}\sqrt{\Delta t}, \end{array}$$(52)$$\begin{array}{ccc} \Delta a_{h} & = & h\left(a_{h},I;C_{a_{h}},C_{I}\right)\Delta t+\xi_{2}\sqrt{\Delta t}, \end{array}$$
and the time evolution is computed by iterating *N* perturbations equally distributed in time with intervals of length Δt. The evolution time, $t_{e}$ (see Figure 5), guarantees that for small perturbations the system is always allowed to return to the steady state. Trajectories of $10^{4}$ steps for the MΩ regime, with $\Delta t=10^{-4}t_{e}$, are obtained. Due to the random nature of these external perturbations, one might expect that the system runs away from the fixed point. However (see below), the size of the perturbations is such that even for extended periods, the system will remain close to the stable state.

In Figure 9, a representative case of a stochastic trajectory is depicted. Two trajectories are displayed, one considering the restitution forces (blue) and the other one only the random perturbations without restitution dynamics (pink). As it can be seen, for the particular case depicted, the trajectory with stability effects maintains the system in the zone of thermodynamic success, exhibiting lower values of entropy production and a low loss coefficient (*D*), with a stable and higher efficiency. Despite the good performance shown in Figure 9, a statistical analysis of many trajectories should be made to guarantee that the induced optimization is an observable quantity.

### 7.2. Statistical Convergence

The previous analysis is repeated for $5\times 10^{4}$ trajectories. For each one, final and average states for *P*, η, Ω, and $\tilde{\sigma }$ are computed. To check for statistical convergence of the results, the Kullback–Leibler divergence, $D_{KL}$, of the distribution of the power output is computed each time that $10^{3}$ trajectories are calculated (the same could be performed for the other thermodynamical functions). This provides a measure of how distant one distribution is compared with the previous one. If $D_{KL}=0$, the information stemming from both distributions is the same and the results obtained are statistically representative.

The interval between the largest and smallest *p* values is divided by $\sqrt{N}$ (rounded to the upper next integer) equal intervals or bins; in this way, the same partition is used to compute the discrete probability distributions of the first *k*-thousand trajectories, $\rho_{k}$ are obtained, and the $D_{KL}$ is calculated comparing $\rho_{k-1}$ with $\rho_{k}$.$D_{KL,k}$.(53)$$D_{KL,k}\left(\rho_{k-1}∥\rho_{k}\right)=-\sum_{i}\rho_{k-1,i}\;log\left(\frac{\rho_{k,i}}{\rho_{k-1,i}}\right),$$
giving a measure of how much information is gained by adding more trajectories. In Figure 10a, the resulting relative entropy $D_{KL}$ is shown. In Figure 10b, it is possible to see that from $2\times 10^{4}$ trajectories the statistical behavior does not vary significantly and adding trajectories will not provide further information. In Figure 10c, the difference between $D_{KL}$ and its previous value is depicted. The difference is small enough to consider that 50 $\times 10^{5}$ trajectories are enough to analyze the stability phenomena at hand.

### 7.3. Effect of Stability on Many Trajectories

For each of the $5\times 10^{4}$ trajectories, the final state of the system is depicted in Figure 11 for the three cases of dynamics: where the influence of *I* is larger, for the symmetrical case, and when $a_{h}$ dominates the stability dynamics (blue points), and they are compared with the cases in which there is no stability dynamics (ND, in pink). Each trajectory consists of $10^{4}$ steps. Notice that in every case the system remains inside the region of thermodynamic success (Δ>0 or equivalently $\eta >\eta_{C}/2$), and especially for the cases (a) and (c), the system is more concentrated in that region. The mean final states are indicated with a blue ⊗ symbol for the ND case and with a black ⊙ symbol for the case with dynamics.

In the ND case, the average final state is displaced to a slightly lower value of *I* (less endoreversible) and at a larger $a_{h}$ (in the direction of the MP state and to the boundary of the success region). However, in the presence of restitution dynamics, the average final state remains at the MΩ state corresponding to a slightly lower value of *I* (displaced over the green line but towards a less endoreversible state) if it is strongly influenced by $a_{h}$. Otherwise, the average final state remains located at the MΩ corresponding to a higher *I* value (displaced over the green line but towards a more endoreversible configuration).

The average behavior of each trajectory is calculated with the mean values of the energetic functions: power output, efficiency, and entropy production. The average behavior of the loss parameters $D_{e}$, $D_{i}$, and *D* are calculated as well. The results for all the trajectories and the three restitution coefficients’ cases are shown in Figure 12. The first row corresponds to the case where the dynamics are dominated by *I*, the second row is the symmetrical case, and the third one is for the case where $a_{h}$ dominates the dynamics.

In the part of the energetic functions, it is possible to see that the first two rows exhibit dynamics where the system tends to evolve in two directions, first, staying between the maximum efficiency and maximum Omega functions, and second, by increasing the power output without big drops in the efficiency; in some cases the power output even surpasses the MP achievable for the value *I* of the steady state but with a noticeable lower entropy production (see the larger values of the dynamics compared with the red dot of the MP state). This is only possible by increasing the parameter *I* in the phase space $a_{h}$-*I*, in which case the system is more endoreversible. Notice that the total losses parameter *D* varies mostly due to the variation in the external losses ($D_{e}$), effectively avoiding getting close to the boundary of the success region.

For the third row, the behavior is different. The averaged states focus mostly on enhancing the efficiency with smaller variations of power output and entropy production, exhibiting a narrow distribution of points with small variations of *P* and σ. This is achieved while maintaining an almost constant value of $D_{e}$, showing a very flat distribution in this direction. Thus, the variations in the total losses, *D*, come from the variations in the parameter *I*, which is in agreement with Figure 11c.

## 8. Summary and Concluding Remarks

The endoreversible model, proposed 50 years ago, is a phenomenological model that has provided unified and general features related to operation regimes with results relevant in quite different schemes. A noteworthy issue raised in the paper is the analysis of the named thermodynamic region of success for the irreversible Carnot-like heat engine. This region is bounded by considering the distance between the totally reversible and the totally dissipative configurations (both configurations offer the most stable configurations for the heat reservoirs); in between, entropy generation due to the correlations in the exchanges between the reservoirs and the working fluid introduces instabilities. As addressed in the paper, it is possible to define a distance between both extreme states. Regardless of the chosen metric, the inflection point at which both states are at the minimum distance defines the boundary from which the reversibility or irreversibilities dominate. The region where reversibilities dominate (success) is found to be preferred by the stability of the engine.

In the literature, the role of stability as another ingredient in the optimization of heat engines has been discussed, even suggesting that the stability mechanisms can provide a self-optimization mechanism to improve the performance of a heat engine under the influence of external noise and limited control. The proposal for the stability dynamics is justified as follows: the cornerstone of the model is the endoreversible hypothesis, which relies on local equilibrium to produce the effective isotherms which in the model appear as the auxiliary reservoirs. In this work, the stability analysis is made on two components; one is related to the variable $a_{h}$ that establishes the operation regime, and the other component is the capacity of the system to produce the effective isotherms to fulfill the endoreversible hypothesis. The parameter *I* provides a measure of how reversible the effective Carnot-like cycle described by the working fluid is (how close the system is to fulfilling the Clausius equality in the inner cycle).

The analysis has been made by selecting the maximum ecological or maximum Omega operation regime, which is a compromise between maximum power and minimum power losses or entropy released to the ambient surroundings. The proposed stability comes solely from recognizing that there exists a steady state and that the stability is well described by the linear approximation (which at least is guaranteed in small perturbations). The resulting dynamics show that restitution velocity favors a slow evolution in the regions between the maximum efficiency and the maximum ecological states. Depending on the direction in which the dynamics is stronger ($a_{h}$ or *I*), the slow region shape can vary, but in all cases the system tends to evolve rapidly to enter into the so-called success thermodynamic region and evolve slowly once inside of this success region. The evolution of the system under random perturbations clearly shows that, without the presence of the stability dynamics, the system arbitrarily evolves inside or outside the thermodynamic success region, which significantly decreases the performance of the engine in power output, efficiency, and entropy generation. On the other hand, when the stability dynamics is acting, the system effectively remains inside the success region. The analysis of many stochastic trajectories confirms this fact and shows that if the thermalization dynamics is strong (dynamics mostly dominated by *I*), the net effect is to produce a slight increase in the reversibility of the system. On the other hand, if the control on the operation parameter $a_{h}$ is strong, the efficiency is enhanced, the value of *I* is slightly decreased, and changes in the total losses are linked to internal losses. The results obtained are statistically representative. This is confirmed by the analysis of the Kullback–Leibler divergence, which shows that the system’s statistical behavior does not change from around 30 thousand trajectories; here, 50 thousand trajectories are computed.

The stability proposed does not consider specific characteristics of the machinery or technology involved in the engine operation and does not include optimization mechanisms. Nonetheless, a link between optimization and stability is found, strengthening the basic idea that stability is another ingredient to consider in the optimization analysis of heat engines. It would be of interest to incorporate a realistic weak control in the simulations of systems such as the one appearing in [12]. If the role of stability and the tendency of local equilibrium to stabilize the system can be generalized to other models, then the local equilibrium can be seen as a mechanism that optimizes and promotes the evolution of energy converters in natural and artificial heat engines.

## Figures and Tables

**Figure 1 entropy-27-00822-f001:**
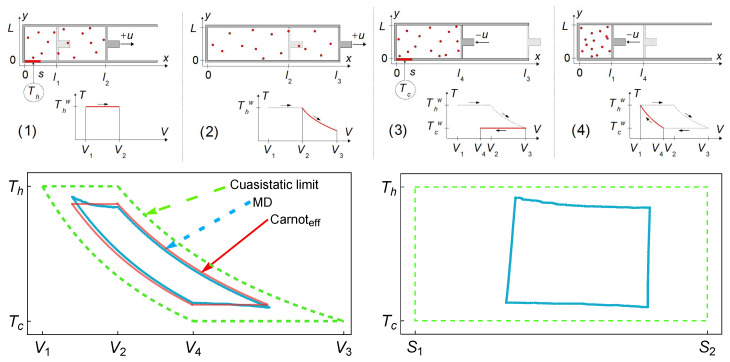
**Upper row**: Schematic diagram of the finite-time Carnot engine (2D) extracted from Figure 1 of Ref. [12]. **Lower row**: Results extracted from Figures 5 and 6 of Ref. [12] stemming from the molecular dynamics (MD) simulations in the Temperature–Volume (T-V) and Temperature–Entropy (T-S) diagrams. The green dashed lines correspond to the reversible quasistatic cycle, and cyan dotted cycles correspond to the MD results and are the average of 2000 cycles under steady state and maximum power conditions. The red cycle represents the effective Carnot cycle, equivalent to the MD results, with *τ* = 0.5.

**Figure 2 entropy-27-00822-f002:**
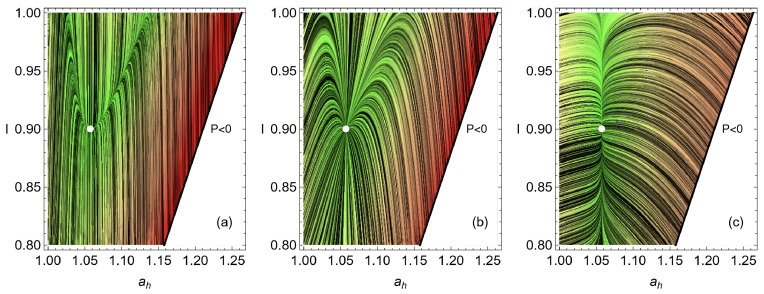
Line integral convolution plot showing streamlines of fixed arc length over a set of random conditions for the dynamics given by solving Equations (Equation 47) and (Equation 48) for the values $\sigma_{hc}=1.4$, $\sigma_{ih}=0.2$, $I_{MP}=0.9$, τ=0.5 and for the case (**a**) $C_{I}=1$ and $C_{a_{h}}=10$, in (**b**) the symmetrical case $C_{I}=C_{a_{h}}=1$ and in (**c**) $C_{I}=10$ and $C_{a_{h}}=1$. The white dot in the center corresponds to the MP operation state, and the white region on the right side of each figure is the nonphysical region where P<0.

**Figure 3 entropy-27-00822-f003:**
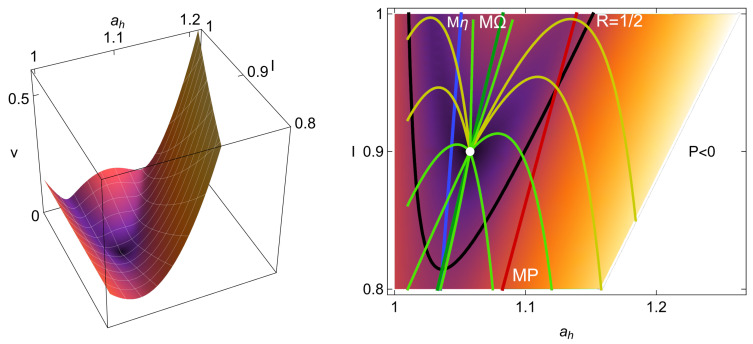
The influence of the velocity of the dynamics (**left**) for the configuration depicted in Figure 2b ($\sigma_{hc}=1.4$, $\sigma_{ih}=0.2$, $I_{MP}=0.9$, τ=0.5 and the symmetrical case $C_{I}=C_{a_{h}}=1$) and some representative relaxation trajectories (**right**). The dashed lines indicate the possible states of maximum power (MP, in red), the maximum ecological and Omega functions (MΩ, in green), and the maximum efficiency (Mη, in blue). The black curve for which R=1/2 indicates the boundary of the success region discussed in Section 3.

**Figure 4 entropy-27-00822-f004:**
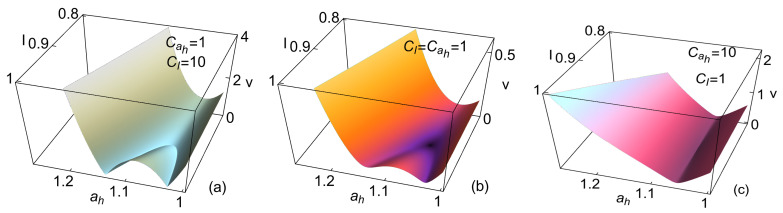
Relaxation velocity for when (**a**) the influence of *I* is larger, $C_{I}=10$ and $C_{a_{h}}=1$. In (**b**), the symmetrical case $C_{I}=C_{a_{h}}=1$. In (**c**), $a_{h}$ dominates the stability dynamics, $C_{I}=1$ and $C_{a_{h}}=10$. In all the cases, $\sigma_{hc}=1.4$, $\sigma_{ih}=0.2$, $I_{MP}=0.9$, and τ=0.5.

**Figure 5 entropy-27-00822-f005:**
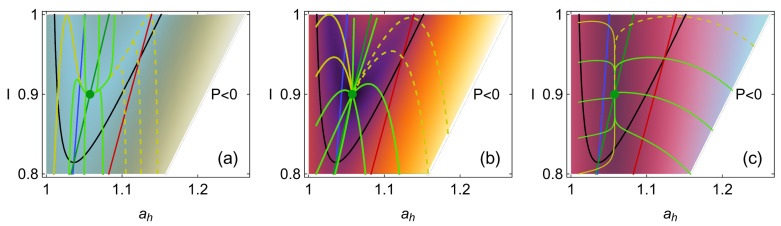
Relaxation trajectories for the cases depicted in Figure 4. The evolution time, $t_{e}$, in each case is (**a**) $t_{e}=28t_{relax}$, (**b**) $t_{e}=11t_{relax}$, and (**c**) $t_{e}=22t_{relax}$.

**Figure 6 entropy-27-00822-f006:**
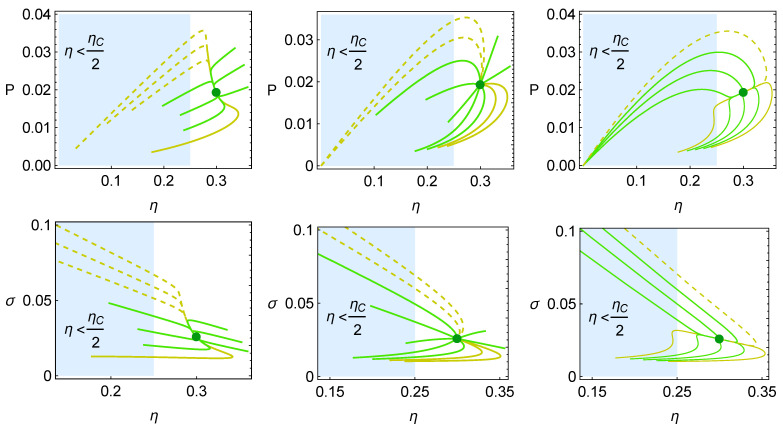
Evolution of σ and η in the relaxation trajectories towards the steady state. The cases correspond to the configurations depicted in Figure 5.

**Figure 7 entropy-27-00822-f007:**
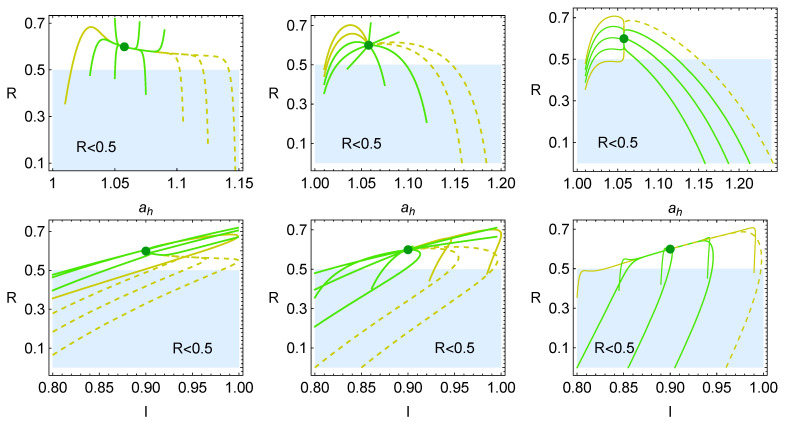
Evolution of the reversible coefficient, *R*, in relaxation trajectories in terms of $a_{h}$ (**upper row**) and *I* (**lower row**) for the cases depicted in Figure 5.

**Figure 8 entropy-27-00822-f008:**
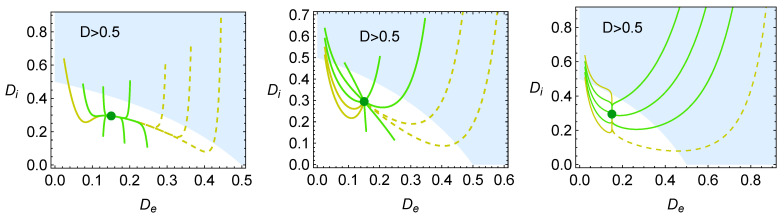
Evolution of internal and external irreversible parameters $D_{i}$ and $D_{e}$ in the relaxation of the system towards the steady state. The three cases depicted correspond to those appearing in Figure 4.

**Figure 9 entropy-27-00822-f009:**
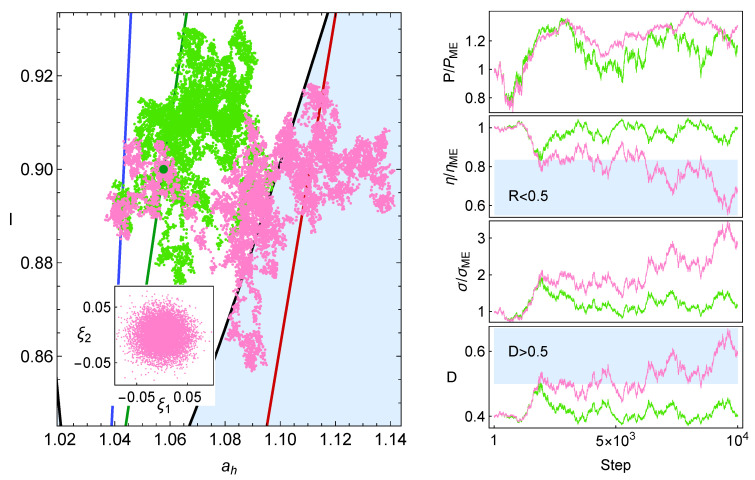
*To the left:* Two representative stochastic trajectories under the influence of the same random perturbations with $10^{4}$ steps. One considers the restitution forces (blue) and the other one does not have restitution dynamics (pink). The random perturbations in the two directions are indicated in the close caption. *To the right:* The time series of *P*, η, and σ (the three of them normalized to the value of the steady state, MΩ), along with the loss coefficient *D*. In the time series of the efficiency and in *D*, the region with R>0.5 (no thermodynamic success) is shaded in blue.

**Figure 10 entropy-27-00822-f010:**
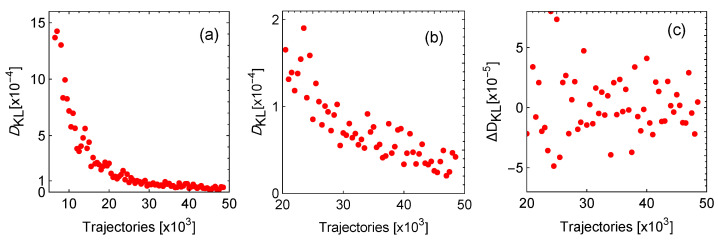
In (**a**) is the Kullback–Leibler divergence, $D_{KL}$, for the power output distribution of every one thousand trajectories. In (**b**) is a close-up for starting at 20 k trajectories, and in (**c**) is the difference between $D_{KL}$ from one point to the next to show that the series is converging.

**Figure 11 entropy-27-00822-f011:**
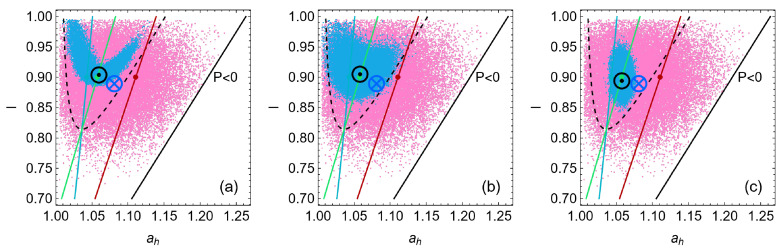
Final states after every stochastic trajectory ($5\times 10^{4}$ trajectories, each one consisting of $10^{4}$ steps). The points in pink correspond to the case where there are no stability dynamics, only random perturbations in the direction of *I* and $a_{h}$. Blue points correspond to the final states in the presence of restitution dynamics. The three cases depicted correspond to those appearing in Figure 4 ($\sigma_{hc}=1.4$, $\sigma_{ih}=0.2$, $I_{MP}=0.9$, and τ=0.5); in (**a**) the influence of *I* is larger $C_{I}=10$ and $C_{a_{h}}=1$. In (**b**), the symmetrical case $C_{I}=C_{a_{h}}=1$. In (**c**), $a_{h}$ dominates the stability dynamics, $C_{I}=1$ and $C_{a_{h}}=10$. The black boundary on the right delimits the region where P>0 and the dashed black curve delimits the region of success. The states of maximum power, maximum efficiency, and maximum ecological and Omega functions are indicated as in Figure 3; the green line corresponds to the MΩ regime, the red line to the MP, and the blue one to M*η*, and the points in each line indicate the operation regime with the *I* value of the steady state.

**Figure 12 entropy-27-00822-f012:**
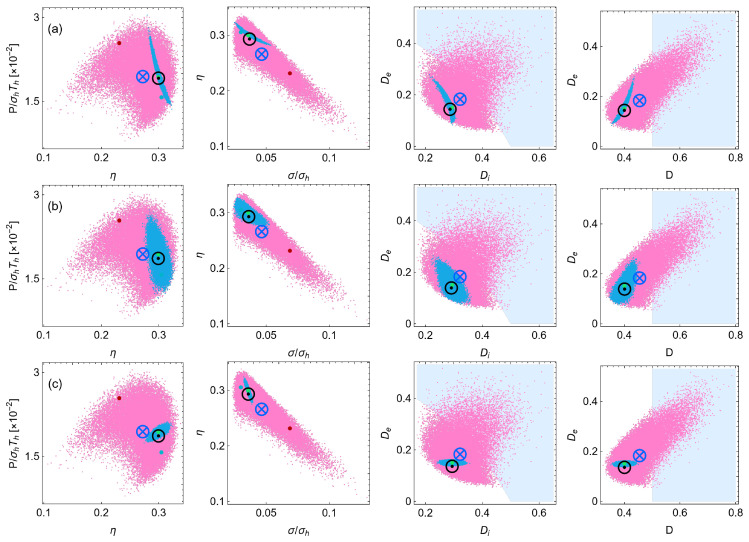
The three cases (**a**–**c**) depicted correspond to those appearing in Figure 4.

## Data Availability

The raw data supporting the conclusions of this article will be made available by the authors on request.

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
