# Peer review of "Linking Optimization Success and Stability of Finite-Time Thermodynamics Heat Engines"

_entropy, 2025, doi:10.3390/e27080822_

Round 1
Reviewer 1 Report
Comments and Suggestions for Authors
In this paper, the stability region for irreversible Carnot-like heat engines assuming endoreversibility is investigated. It is theoretically and numerically shown that more reversible (success) region is preferred by the engine from the dynamical stability point of view.
Although this work may be based on the previous works by the authors such as Refs.[84,86,87,88], extending them to the endoreversible regimes adjusted with an internal irreversibility parameter, it clarifies the relationship between optimization and stability in irreversible Carnot-like heat engines and would contribute to the field of finite time thermodynamics. Therefore, I could recommend this work for publication after the authors consider the following points.
First, the presentation of the overall manuscript should be improved more carefully. For example, probably, the second equality of Eq.(41) would be imperfectly defined. Moreover, the relationship between the parameter $I$ and the correlation entropy is unclear. Please discuss it in more detail.
As the second point, I believe the stability dynamics Eq.(45) is an assumption imposed on this model, which may not necessarily be derived from other (more basic) equations. But, in practice, what determines the restitution strength? What is the meaning of the parameter $t$ ? (Is it a time?) Discussion on this point would be helpful for readers to understand the physics behind the stability dynamics.
Author Response
Comment 1: In this paper, the stability region for irreversible Carnot-like heat engines assuming endoreversibility is investigated. It is theoretically and numerically shown that more reversible (success) region is preferred by the engine from the dynamical stability point of view.
Although this work may be based on the previous works by the authors such as Refs.[84,86,87,88], extending them to the endoreversible regimes adjusted with an internal irreversibility parameter, it clarifies the relationship between optimization and stability in irreversible Carnot-like heat engines and would contribute to the field of finite time thermodynamics. Therefore, I could recommend this work for publication after the authors consider the following points.
Answer: We appreciate the comments made by the Reviewer, which helped us to improve the manuscript and explain in a clearer manner some subtleties regarding the dynamical equations, the role of the irreversible parameter I, and some other quantities introduced in the manuscript.
Comment 2: First, the presentation of the overall manuscript should be improved more carefully. For example, probably, the second equality of Eq.(41) would be imperfectly defined. Moreover, the relationship between the parameter $I$ and the correlation entropy is unclear. Please discuss it in more detail.
Answer: We have improved the definition of the parameter I and explained in a more concise manner the rationale for its use, so readers from other areas of thermodynamics and physics have a better comprehension of it. The relationship between the correlation entropy and the parameter I, as the Reviewer points out, needed a more concise explanation. This is addressed more carefully in the text. Section 4 and the beginning of Section 5 have been modified for this purpose.
Comment 3: As the second point, I believe the stability dynamics Eq.(45) is an assumption imposed on this model, which may not necessarily be derived from other (more basic) equations. But, in practice, what determines the restitution strength? What is the meaning of the parameter $t$ ? (Is it a time?) Discussion on this point would be helpful for readers to understand the physics behind the stability dynamics.
Answer: Yes, the equilibrium in this equation comes from the assumption that there exists an equilibrium mechanism, which has been somewhat justified in section 1.3 (in the revised version we have improved the discussion about this issue) by analyzing a more realistic simulation based on kinematic interactions. The thermalization achieved by the working fluid reproducing almost thermal equilibrium reinforces the assumption that the time scale of the interactions is small enough so local equilibrium can bring the system near equilibrium, which is a stable state. This idea is explained in more detail in the text (red text in Section 5), explaining the lack of information at hand for determining the restitution strengths, and the analysis of three scenarios is made for that reason.
Reviewer 2 Report
Comments and Suggestions for Authors
The paper is dedicated to the very interesting problem of stability of finite-time heat engines and its relation to the operating modes of such an engine. The language of stability theory is, unfortunately, not widely spread within the FTT community, so I appreciate the effort made by the authors and think that the paper has some inherent value.
I have the following comments for the authors:
1. English language could and must be improved. There are places where the language is almost perfect and there are places which meaning is somewhat obscure.
2. Reference list is huge but does not contain some important references. For example there are no references to the historical papers by Feidt (which I think are crucial when talking about the history of FTT in the Introduction) or the papers on stability in thermodynamics by Andresen (or even Prigogine works). The only papers in the Reference list related to thermodynamic stability are written by the authors. Some papers by Tsirlin and colleagues (those dedicated to distillation for example) contain the notions of the working and non-working regions in the reachable set of some process. Are these regions the same as the success and failure regions in the manuscript under review?
3. There are too many references to microscopic frameworks, but none of them is being used in the paper and, in my opinion, none of them should be used when talking about endoreversible models of a heat engine.
4. Authors introduce a lot of quantities not very common in the field of thermodynamics (for example, I, a_h, \tau, R, D, \Delta, etc). The physical meaning of these quantities is unclear (e.g. what does dI/dt mean?). I understand that this is the established notation used by the authors, but I think that they should present it in some form for the convenience of the reader.
5. While the overall idea of the paper is described rather clear in the abstract, the manuscript does not contain clear formulations of the problems being solved. Introduction of these formulations will improve the manuscript significantly.
I see these comments as minor ones and recommend publishing this paper after addressing some of the issues.
Author Response
Comment 1: The paper is dedicated to the very interesting problem of stability of finite-time heat engines and its relation to the operating modes of such an engine. The language of stability theory is, unfortunately, not widely spread within the FTT community, so I appreciate the effort made by the authors and think that the paper has some inherent value.
I have the following comments for the authors:
1. English language could and must be improved. There are places where the language is almost perfect and there are places which meaning is somewhat obscure.
Answer: We thank the Reviewer for the recommendations. We have improved the text along the manuscript; some concepts have been explained in more detail, the interpretation of several quantities, and the use of the proposed dynamics is explained in more detail as well. Some parameters were eliminated, as their definition is not necessary for the discussion, such as the parameter Delta=R-D, which still can be analyzed.
Comment 2: Reference list is huge but does not contain some important references. For example there are no references to the historical papers by Feidt (which I think are crucial when talking about the history of FTT in the Introduction) or the papers on stability in thermodynamics by Andresen (or even Prigogine works). The only papers in the Reference list related to thermodynamic stability are written by the authors. Some papers by Tsirlin and colleagues (those dedicated to distillation for example) contain the notions of the working and non-working regions in the reachable set of some process. Are these regions the same as the success and failure regions in the manuscript under review?
Answer: We are grateful to the referee for mentioning the historical work on FTT by several authors in FTT. New references have been added, but the subtle variations, such as FST (Finite Speed Thermodynamics), or FDOT (Finite physical Dimensions Optimal Thermodynamics) are not discussed in the paper to not distract the reader from the main issue (which is the major concern in the introduction) of justifying the definition of the stability dynamics. We also included references to the work by Andresen and Tsirlin concerning the tools for analyzing thermodynamic systems involving time or rate constraints as the extended thermodynamic potentials and the conditions of optimal regimes based on minimum entropy production and availability using the optimal control theory from balance equations for energy, entropy, and mass. This has been englobed in a brief comment at the end of Section 1.2. The new references are [79-83]. Regarding the previous works by Andresen and Tsirlin, in their generic framework, the entropy production is minimum under size constraints and flow intensities, so that its minimum value finally depends on a function of the flow intensities and the coefficients of the kinetic equations. The resulting "region of feasible conditions" is finally dependent on the kinetic equations. These ingredients are absent in our stability formalism, which in turn rests on the restitution forces and where the entropy is not bounded at all. Then, in our opinion, "success and failure regions" (in our work) are hardly comparable with "region of feasible conditions".
Comment 3: There are too many references to microscopic frameworks, but none of them is being used in the paper and, in my opinion, none of them should be used when talking about endoreversible models of a heat engine.
Answer: We agree that some references are not closely related to the model and the topic at hand. Thus, we have made a double-check for the references that should be included and excluded from the manuscript.
Comment 4: Authors introduce a lot of quantities not very common in the field of thermodynamics (for example, I, a_h, \tau, R, D, \Delta, etc). The physical meaning of these quantities is unclear (e.g. what does dI/dt mean?). I understand that this is the established notation used by the authors, but I think that they should present it in some form for the convenience of the reader.
Answer: We have been more explicit in the definition of the dynamical equations and several parts of the manuscript have been improve to better explain the parameters used. These changes are marked in red color along the manuscript.
We have improved the definition of the new parameters and we eliminated \Delta from the text, since it was not necessary for the discussion, see for example the red text around eqs. (26)-(31). Also, Section 4 presents several changes to improve the interpretation of the correlation entropy and the parameter I. Also, the dynamical equations are introduced in a more detailed manner.
Comment 5: While the overall idea of the paper is described rather clear in the abstract, the manuscript does not contain clear formulations of the problems being solved. Introduction of these formulations will improve the manuscript significantly.
Answer: We agree that a more careful explanation of some quantities was needed; we hope that the changes made to the manuscript make the text more readable.
Recommendation: I see these comments as minor ones and recommend publishing this paper after addressing some of the issues.
Answer: Once again we thank the Reviewer for the careful reading and the recommendations.